# Bone Formation and Maintenance in Oral Surgery: The Decisive Role of the Immune System—A Narrative Review of Mechanisms and Solutions

**DOI:** 10.3390/bioengineering11020191

**Published:** 2024-02-16

**Authors:** Elisa Choukroun, Maximilien Parnot, Jerome Surmenian, Reinhard Gruber, Nicolas Cohen, Nicolas Davido, Alain Simonpieri, Charles Savoldelli, Franck Afota, Hicham El Mjabber, Joseph Choukroun

**Affiliations:** 1Private Practice, 06700 Saint Laurent du Var, France; dr.mparnot@gmail.com; 2Private Practice, 06000 Nice, France; jerome.surmenian@me.com (J.S.); afota.f@chu-nice.fr (F.A.); 3Department of Oral Biology, University Clinic of Dentistry, Medical University of Vienna, 1090 Vienna, Austria; reinhard.gruber@meduniwien.ac.at; 4Private Practice, Kirkland, ON H9J 2K9, Canada; drnicolascohen@gmail.com; 5Private Practice, 75017 Paris, France; nicolas.davido@gmail.com; 6Private Practice, 06240 Beausoleil, France; 7Head and Neck Institute, CHU, 06000 Nice, France; csavoldelli@yahoo.fr; 8Private Practice, 42600 Montbrison, France; hichmail@hotmail.fr; 9Pain Clinic, 06000 Nice, France

**Keywords:** osteoimmunology, oxidative stress, peri-implantitis, bone regeneration, bone graft, implant dentistry, immune system, oxidants, antioxidants

## Abstract

Based on the evidence of a significant communication and connection pathway between the bone and immune systems, a new science has emerged: osteoimmunology. Indeed, the immune system has a considerable impact on bone health and diseases, as well as on bone formation during grafts and its stability over time. Chronic inflammation induces the excessive production of oxidants. An imbalance between the levels of oxidants and antioxidants is called oxidative stress. This physio-pathological state causes both molecular and cellular damage, which leads to DNA alterations, genetic mutations and cell apoptosis, and thus, impaired immunity followed by delayed or compromised wound healing. Oxidative stress levels experienced by the body affect bone regeneration and maintenance around teeth and dental implants. As the immune system and bone remodeling are interconnected, bone loss is a consequence of immune dysregulation. Therefore, oral tissue deficiencies such as periodontitis and peri-implantitis should be regarded as immune diseases. Bone management strategies should include both biological and surgical solutions. These protocols tend to improve immunity through antioxidant production to enhance bone formation and prevent bone loss. This narrative review aims to highlight the relationship between inflammation, oxidation, immunity and bone health in the oral cavity. It intends to help clinicians to detect high-risk situations in oral surgery and to propose biological and clinical solutions that will enhance patients’ immune responses and surgical treatment outcomes.

## 1. Introduction

In 1594, French anatomist Barthelemy Cabrol, in his publication *Alphabet Anatomic*, was the first to use the term “osteology” to describe the different mechanisms of bone physiology, including those involved in bone formation. In 2000, the concept of “osteoimmunology” was introduced by Arron and Choi [1]. Immune cells and factors are key regulators of wound healing and contribute to the early stages of angiogenesis [2,3]. Increasing evidence suggests that they are involved in bone synthesis and remodeling cascades, through which the immune system controls bone formation, regulates bone resorption and acts as a key factor in bone homeostasis [4,5]. Inversely, immune cell functions are influenced by the bone system [6]. It has been concluded that the bone and immune systems are unified in a single entity: the osteoimmune system [7].

The term “oxidative stress” first appeared in the medical literature in 1985 [8]. Aerobic species, when in contact with oxygen, physiologically produce various types of oxidants. To neutralize them, cells produce antioxidants. However, when the level of oxidants exceeds that of antioxidants in the body or tissues, oxidative stress occurs. This physio-pathological state can cause both molecular and cellular damage, such as DNA alterations, genetic mutations and apoptosis. Oxidative stress increases the occurrence rates of certain illnesses, accelerates cell aging and prevents the immune system from functioning as it should [9].

A consequence of an impaired immune response and oxidative stress is the early or late failure of osseointegration [10]. Indeed, dental implants have a medium success rate of up to 82.9% after 16 years [11], but inflammatory complications, such as mucositis and peri-implantitis, have been increasingly reported. Mucositis has been defined as peri-implant bleeding on probing, erythema and swelling and/or suppuration, while peri-implantitis is characterized by inflammation in the surrounding mucosa and subsequent progressive bone loss. Systematic reviews estimate the prevalence of peri-implantitis to be 9.0% to 12.8% [12,13,14], and this figure increases with time (26% after five years of function and 21.2% after 10 years) [15].

A lack of regular maintenance, a history of periodontitis and risk factors such as diabetes and smoking have been cited as possible causes of peri-implant bone loss, but the elimination of such risk factors has been only partially preventive. The underlying causes of tissue deficiencies and bone loss around oral implants remain unclear, and various theories, such as bacteriological invasion, immunological reactions and titanium allergies, have been proposed, but without worldwide approval [16].

The available literature on the subject is mainly focused on prosthodontics aspects, but none have given a biological explanation. No evidence has been found that primary infection induces marginal bone resorption [17].

From an osteoimmune point of view, success in dental implants or bone augmentation procedures could depend on the presence of sufficient antioxidants and, thus, on an effective immune response. Bone loss could be prevented through the management of immunity and oxidation before, during and after surgery.

Through this narrative review, the authors aspire to guide oral surgeons in the comprehension of osteoimmunology and oxidative stress mechanisms and, based on these principles, to introduce concrete clinical protocols by which to prevent bone augmentations failures and peri-implantitis.

## 2. Osteoimmunology

Arron and Choi first introduced the concept of osteoimmunology in 2000 [1], based on the strong connections that are found between the bone and immune systems [18].

### 2.1. Immune System and Immunity

The immune system is a network of organs that comprise white blood cells, proteins (antibodies) and chemicals, working together to defend against foreign invaders such as bacteria, viruses, parasites and fungi, which can cause infection, illness and diseases [19]. Immunity involves the scavenging of microorganisms, exogenous materials, abnormal endogenous materials, waste products and diseased cells. Weakened immune functions make the body more susceptible to infections or to the development of malignant tumors.

The immune system consists of two separate subsystems, known as innate (non-specific) and adaptive (specific) immunity. They are closely connected and work in conjunction whenever a germ or harmful substance triggers an immune response.

**a.** **The innate immune system** is the first line of defense. It is the first structure of the body to detect pathogens such as viruses, bacteria, parasites and toxins, or to sense wounds and trauma. Besides playing this role, the innate immune system activates corresponding cells to attack and knock down microbes and initiate tissue healing and remodeling. Inflammation is the main mechanism of innate immunity, and its effector cells are phagocytic cells (granulocytes, monocytes, macrophages, dendritic cells), epithelial and endothelial cells, natural killer (NK) cells, innate lymphoid cells (ILC) and platelets [20]. After this first level of immune response has been activated, mediators inform and modulate the adaptive immune response.**b.** **The adaptive (or acquired) immune system** is the second and specific line of defense and is developed after exposure to microbes or their released chemicals. Lymphocytes are the main cells of acquired immunity. T-lymphocytes are in control of cell-mediated immunity, while B-lymphocytes handle humoral immunity with antibodies. Excessive levels of specific immune responses lead to allergic reactions or the development of autoimmune diseases [21].

### 2.2. Bone Repair Process and Inflammation

Immune and bone cells have a conjoined heritage in stem cells; they share signaling pathways and influence each other permanently [22]. The immune system controls bone and its resorption and thus acts as a key factor in bone homeostasis; a controlled inflammatory immune response is crucial for bone formation, osseointegration and successful regeneration [4,5].

Both innate and adaptive systems contribute to bone homeostasis. After a bone fracture, immune cells, especially macrophages, are found throughout the healing process, as they organize the body’s defenses against pathogens and discharge a complex variety of effectors to regulate bone remodeling [23]. Bone healing starts through an inflammatory reaction, which is a common response of living, vascularized tissue to aggression. Inflammation is induced by the release of leukocyte inflammatory cytokines, predominantly angiogenic [24]. After angiogenesis onset, a granulation tissue, also named “soft bone callus”, is formed [25]. However, two conditions are required for this cascade of events to occur: a sufficient amount of antioxidants and the correct activation of repair genes [26].

The main activator of repair genes is Nuclear Factor Erythroid-2-Related Factor 2 (Nrf2), which is a transcription factor that regulates over 250 genes in the human genome [27]. These genes mainly control damage repair processes and reduce inflammation by increasing antioxidant production. They share a common regulatory sequence named ARE (Antioxidant Response Element).

In normal, non-stressed circumstances, Nrf2 is expressed sparingly inside the cytoplasm. Under stressed conditions or with specific targeted activators, Nrf2 is activated, and then, translocated to the nucleus where it binds to RNA and enhances the transcription of genes for antioxidant enzymes and cytoprotective proteins [28,29]. Nrf2 activation leads to reductions in levels of T cells and macrophages, the anti-inflammatory shift of B-cells and cytoprotective cellular responses.

Inflammation is always beneficial during the first days after damage. However, prolonged inflammation becomes harmful as it results in excessive free radical production and the release of the Receptor Activator of the Nuclear Factor Kappa Beta Ligand (RANKL) from osteoblasts and osteocytes. Long-term inflammation induces continuous bone loss around teeth or dental implants [30,31,32].

### 2.3. Influence of Immune System on Alveolar Bone

Osteoimmunology has already provided valuable insights into periodontitis pathogenesis and the strong immunoreactivity of periodontal tissues [33,34,35] and may offer a new perspective to understand peri-implantitis. The immune response initiated by the dysbiotic microbiota during periodontitis enhances the production of RANKL by different immune cells. Osteoblasts and periodontal ligament cells also respond to IL-17, producing RANKL, and decreasing osteoprotegerin (OPG) production [36].

Periodontitis and peri-implantitis present similar symptoms: chronic inflammation of the periodontal tissue and subsequent destruction of the alveolar bone around teeth or implants [37]. The role of the macrophage–osteoclast axis in bone damage is now well documented: bone loss is caused by inflammation and immune diseases [38]. There is strong evidence that the interleukins IL-1β, IL-6 and IL-10 and RANKL with OPG, which inhibits bone resorption and modulates inflammatory responses, are involved in the pathophysiology of peri-implant diseases [39]. T cells have been shown to have capacities to activate osteoclasts and are thus thought to be responsible for bone loss. In addition, the literature is providing increasing awareness of the role of osteoblasts in the onset of bone maintenance failures through their antigen-presenting properties [36,40].

Numerous clinical scenarios can be envisioned whereby osteoimmune equilibrium is disturbed and marginal bone loss develops to worsen into peri-implantitis [41]. Thus, peri-implantitis should be considered as an immune disease.

## 3. Oxidation and Bone

Oxidation–reduction or redox reactions occur constantly in living creatures, mainly inside the mitochondria and through the NADPH Oxidase enzyme family (Nicotinamide Adenine Dinucleotide Phosphate Oxidases or NOXs).

NOX proteins are principally located in the plasma membrane, but they can also be found on other membranes such as the endoplasmic reticulum and mitochondria. Their main function is to transfer electrons to molecules. The result is the continuous production of free radicals, also named oxidants or reactive oxygen species (ROS). ROS are generated as by-products of aerobic metabolism during oxidative phosphorylation. They are highly reactive molecules after leakage from their outer shell of a single impaired electron.

They comprise several diverse chemical species, and the major forms are the superoxide anion of oxygen (O_2_-), hydrogen peroxide (H_2_O_2_) and hydroxyl radicals (·OH) [42,43,44,45,46].

At low concentrations, ROS are beneficial as they serve as signaling molecules that activate specific physiological pathways controlling numerous life processes, including anti-microbial activity [47,48]. However, they also interact with other molecules to generate secondary ROS, which are more unstable and thus react more aggressively than primary ROS. Secondary ROS interfere with the structures and functions of bodily molecules, mainly lipids, lipoproteins, proteins and nucleic acids. These reactions lead to two types of damage: molecular, as the reactions alter DNA, and cellular, as they cause mutagenesis and carcinogenesis.

The human body balances the levels of oxidants with antioxidants. Antioxidants are low-molecular-weight molecules that counteract oxidation, either by preventing ROS formation, interrupting oxidation reactions or through scavenging and neutralizing ROS by providing the missing electron.

Antioxidants can be produced endogenously and distributed within the cytoplasm. They are either thiol-based or non-thiol compounds such as polyphenols, vitamins and various enzymes. Among the most important are SuperOxide Dismutase (SOD) and Glutathione Peroxidase (GPx), which convert ROS into stable molecules such as H_2_O and O_2_ and thus limit their side-effects [5]. People can also obtain antioxidants through exogenous oral supplementation.

### 3.1. Oxidative Stress

Oxidative stress occurs when oxidant and antioxidant levels are imbalanced and ROS production overwhelms cellular antioxidant defenses [42,49]. Free radical production, due to elevated NOX activity and deregulation, are significantly increased under several conditions, including tissue damage repair, exposure to pollutants, unbalanced diet, anxiety, smoking and chronic inflammation.

Oxidative damage to bio-macromolecules plays an active role in the etiology of a wide variety of acute and chronic diseases [50]. It plays a central role in the acceleration of aging and osteoporosis [51,52,53]. The aberrant production of free radicals is implicated in numerous diverse pathologies, such as cancer, arthritis and cardiovascular diseases, and may induce uncontrollable autoimmune illnesses [54,55,56,57]. Moreover, due to ROS reactions with brain neurons, oxidative stress is involved in the development of neurodegenerative conditions such as Alzheimer’s and Parkinson’s diseases and other forms of dementia [58].

Oxidative stress can be clinically detected in patients. There are seven well-known signs: increased fatigue; memory loss and/or brain fog; muscle and/or joint pain; wrinkles and grey hair; impaired eyesight; headaches and sensitivity to noise; and susceptibility to infections [59].

### 3.2. Oxidative Stress and Bone

Bone is remodeled in a complex cycle that lasts for approximately six months and involves three main groups of bone cells: osteoclasts, osteoblasts and osteocytes. Under normal circumstances, the cycle is regulated by cytokines, growth factors and hormones [60,61]. Physiological redox changes occur during bone remodeling, and alterations in ROS and/or antioxidant levels can affect bone homeostasis and remodeling [62].

Elevated amounts of free radicals in osteoblasts inhibit their functions [63,64] and lead to the apoptosis of both osteoblasts and osteocytes [65]. Excessive levels of osteocyte apoptosis disrupt the balance in favor of osteoclastogenesis and inhibit the osteogenesis and mineralization processes. This fast cell death increases bone remodeling turnover and favors bone loss.

ROS play a crucial role in osteoclast differentiation and function [57,66]. Oxidative stress promotes the differentiation of pre-osteoclasts into mature osteoclasts while inducing the apoptosis of osteoblasts and osteocytes, thereby increasing rates of bone resorption [67].

At a molecular level, a high level of oxidation increases RANKL production and activates the molecular processes that involve Extracellular Receptor Kinase (ERK), Nuclear Factor Kappa B(NF-κB), Tumor Necrosis Factor (TNFα), InterLeukin-6 (IL-6) and Metalloproteinase 2, 8 and 9 (MMP-2/8/9). This activation promotes osteoblast apoptosis and osteoclastogenesis [68].

Antioxidants, on the other hand, have opposite effects: they enhance mineralization processes and reduce osteoclastic activity, either directly or by counteracting the effects of oxidants [69].

Recent studies have highlighted the crucial role of Nrf2 in osteoporosis, osteogenic stem-cell differentiation and bone fracture healing. For instance, the Nrf2-mediated activation of antioxidant signaling prevents estrogenic osteoporosis [70]. Nrf2 activation facilitates the osteogenic differentiation of bone mesenchymal stem cells [71] and regulates osteoclasts and osteoblasts in bone homeostasis [72]. Conversely, impaired Nrf2 nuclear translocation is associated with delayed cortical bone healing [73].

## 4. Origins of Oxidative Stress and Immune Deficiency in Oral Surgery

Elevated ROS levels, which are associated with metabolic diseases, chronic ischemia or inflammation, compromise immune functions and thus affect bone formation and maintenance (Figure 1).

### 4.1. Biological and Metabolic Effects

Diabetes and smoking induce extended redox reactions, which lead to steep increase in levels of oxidants and therefore cause complications and failures in physiological mechanisms [74,75].

Diabetes, as a chronic inflammatory disease [76], induces oxidative stress through hyperglycemia [77]. The healing issues of diabetics, their systemic complications (including nephropathy, neuropathy and angiopathy) and periodontal fragility are related to their oxidant levels [78,79].

Tobacco smoke, in contact with tissues such as the skin, lungs or those in the oral cavity, breaks down their antioxidants, which leads to local oxidation [80]. Smoke exposure impairs the oxidant/antioxidant balance in oral tissue and increases the release of pro-inflammatory cytokines [81]. The resulting inflammatory response and oxidative stress in peri-implant tissue delay wound healing and increase the risk of peri-implantitis.

Vitamin D is also a major factor in ROS production. Its normal range in serum is 30–100 ng/mL (or 75–250 nmol/mL). Vitamin D is synthetized mainly after skin exposure to UV radiation [82]. It undergoes a double hydroxylation in the kidney and liver. The resulting active hormone, 1,25-(OH)_2_-VitD3, is involved in the endocrine and paracrine pathways. Both enable vitamin D to perform a neuromediative function: it regulates cell growth and stimulates the production of antioxidants [83]. Numerous genes determining immunity and bone growth or remodeling are vitamin D-dependent, and Vitamin D is one of the most powerful activators of Nrf2 [84,85]. It stimulates the expression of anti-microbial peptides that are effective against bacteria such as *Staphylococcus aureus* [86,87]. It has also been shown to have anti-inflammatory properties, as it inhibits Cox-2 expression and suppresses pro-inflammatory mediators [88].

Vitamin D deficiency is widespread, due to modern human lifestyles combining low sun exposure and time mostly spent indoors. Studies suggest that 70–80% of the population may be deficient in vitamin D. A level of vitamin D under 20 ng/mL can lead to systemic inflammation, organ dysfunction, high infection rates and extended lengths of stay in hospital [89,90,91]. Risk factors for deficiency are age, dark skin, depression, obesity, smoking, diabetes, allergies and kidney insufficiency. Vitamin D deficiency significantly impacts osseointegration and the success rates of dental implant therapies and bone graft outcomes [92,93,94,95].

Numerous studies have reported a correlation between low mineral density of bones and high levels of total and low-density-lipoprotein (LDL) cholesterol. LDL cholesterol is oxidized in osteoblasts and is frequently implicated in slow bone metabolism, during which bone synthesis is slowed and stem cells are oriented to produce fat cells [96]. The bone becomes fattier than normal and takes on a yellowish color. In contrast, high-density-lipoprotein (HDL) cholesterol acts as an antioxidant [97,98].

### 4.2. Penicillin Allergy

Prior to oral surgery, antibiotic prophylaxis is usually prescribed to reduce the amount of bacteria at the surgical site. However, prophylaxis can have a limited effect in patients allergic to penicillin [99]. This population has been shown to carry a three- to four-fold increased risk of surgical site infection in comparison with non-allergic patients [100] and alternatives to amoxicillin do not show its levels of efficiency [101]. Allergic patients recruit insufficient neutrophils to ward off pathogens. Allergy is an immune incompetency in which T-lymphocytes have been identified as the main agent responsible for the long-term release of inflammatory cytokines [102]. Moreover, allergic patients are often deficient in vitamin D [103]. Consequently, these patients are in chronic oxidative stress and present an inadequate immune response [104,105].

### 4.3. Chronic Inflammation

Aside from inflammatory diseases (autoimmune diseases, kidney diseases, obesity, diabetes, Human Immunodeficiency Virus (HIV), etc.), oral chronic inflammation may be a result of chronic contamination of the periodontal tissues.

In oral surgery, one of the main concern is the amount of pathogens present in the oral cavity. Normally, the gingival epithelium should act as a barrier to bacteria. However, in many patients, the junctional epithelium is permeable to bacteria [106], and therefore can form a gateway for microorganisms to contaminate the underlying soft tissue. This occurs particularly in patients with a thin and permeable biotype, which represent about one third of adults (about 50% of women and 25% of men) [107]. Pathogens spread to the periodontium and induce long-term inflammation. The resulting oxidative stress induces bone loss and fibrosis if soft tissues around implants is not improved by thickening [108,109]. These kinds of issues can also be found upon re-entry after bone augmentation when the periodontium has been contaminated during the healing period.

### 4.4. Chronic Hypoxia

High levels of oxidants are also produced during chronic hypoxia. Tissues become hypoxic as they lose their vasculature when exposed to overpressure or tension [110].

Blood to the cortical bone is supplied from the soft tissue through the periosteum; therefore, any soft-tissue hypoxia leads to underlying bone hypoxia and condemns it to resorption.

The placement of a dental implant with high primary stability is likely to create bone compression. On cancellous bone, implants, even when placed with high torque, will not induce pressure due to its flexibility. In contrast, the cortical bone is rigid and has a reduced blood supply. Consequently, torque, and thus, pressure, can cause marginal bone loss, especially if the cortical bone is thick [111].

Regarding grafted bone, the bone matrix becomes stiff after a few months. Mechanically, the regenerated bone has low flexibility and behaves as a cortical bone when submitted to pressure.

Inversely, when insufficient pressure is placed on a bone because of a lack of mechanical activity, the production of oxidants is increased, and this leads to bone loss. Astronauts who spend long periods in space risk bone loss due to a lack of pressure [112].

On the other hand, tension creates ischemia in soft tissue [113]. Soft-tissue traction may occur if flap release is inconsistent after bone augmentation, the suture technique used is not adapted or if keratinized attached gingiva is lacking [114,115,116]. Bone graft outcomes rely on primary passive-tension-free wound closure.

## 5. Review of Solutions to Improve Immune Response and Reduce Levels of Oxidative Stress

In the oral surgical field, an improved understanding of osteoimmunology has enabled the proposal of guidelines to enhance bone formation, reduce complications or failures and achieve long-term stability. Improvements in immune responses can be achieved through the use of antioxidation or anti-inflammation strategies (Figure 2).

### 5.1. Systemic Improvements in Immune Response and Anti-Inflammation Strategies

Antioxidation is one way to enhance the immune system and reduce the length of inflammation.

**a.** 
**Antioxidation by nutraceuticals**


The use of external antioxidants or nutraceuticals has been demonstrated to decrease levels of oxidative damage. Oral antioxidants alone have been shown to reduce by up to 40% the levels of blood serum markers of oxidative stress, especially in smokers, a high-risk population [117].

Exogenous antioxidants can be classified into two groups: those with Nrf2 activation functions and those with direct antioxidative properties.


**Antioxidants that activate Nrf2 functions**


It has been proven that Nrf2 activators hold the potential to control inflammation-driven bone loss [60,118]. Nrf2 plays a crucial role in the development of targeted therapies for metabolic bone diseases as well as for bone reconstruction and ossification around implants.


**
*Vitamin D*
**


High level of vitamin D significantly improve the outcomes of oral surgical procedures [92,93,94,95]. The recommended daily supplementation of 400–600 International Units (IU) is effective only in the prevention most skeletal abnormalities [119]. To stimulate autocrine and paracrine pathways and to improve antioxidant levels and cell growth, higher doses of 2000–10,000 IU/day are necessary, depending on pre-op blood test results [120]. Before oral surgery, the patient’s serum level should be upgraded to 50–70 ng/mL (125–175 nmol/mL). Vitamin D blood tests should be performed systematically before any surgical procedure and the patient advised to take supplements in case of deficiency.


**
*Vitamin C or ascorbic acid*
**


Ascorbic acid is a water-soluble antioxidant that functions by giving up an electron to neutralize free radicals. It cannot be synthesized by the human body and, therefore, can only be obtained from exogenous sources. Its tropism for collagen and elastin synthesis of the epithelial barrier enhances wound and connective tissue healing and immunity [121,122]. Vitamin C has been found to help in reducing inflammation and chronic inflammatory disease complications [123]. It modulates osteoblastogenesis and osteoclastogenesis through Nrf2 activation and thereby promotes bone formation, remodeling and healing [124,125]. It can be used to improve healing after oral surgery and to prevent the incidence and development of periodontal disease [126,127,128]. 

At a suggested nutritional intake of 1000 mg/day for adults [129], vitamin C supplementation is highly recommended for the elderly, smokers and diabetics, due to their increased occurrence of inflammatory processes.


**
*Melatonin*
**


Produced by the pineal gland, melatonin is a chronobiotic hormone that regulates the circadian rhythm. Numerous studies have reported its immunomodulatory and immune stimulative functions [130]. Its action on Nrf2 translocation influences both the innate and adaptive immune systems by repressing pro-inflammatory cytokines and reducing oxidative stress [131,132]. Melatonin helps to prevent inflammation, infection, sepsis and immunosenescence.

Melatonin accelerates bone healing and osseointegration and promotes angiogenesis [133]. It attenuates bone resorption and could help to prevent periodontitis and peri-implantitis [134,135].

Melatonin supplementation, ideally at a rate of 1–5 mg/day, should be administered thirty minutes to one hour before bedtime. Several publications have investigated the use of topical melatonin in extraction sockets, mixed with biomaterials or to coat implants. These studies show promising results, but further research is required [136,137].


**
*Zinc*
**


Zinc, as an essential trace mineral element, is a crucial component of key antioxidant enzymes [138] and has demonstrated roles in the modulation of both innate and adaptive immune responses. Its supplementation reduces plasma levels of oxidative stress markers and decreases the production of inflammatory cytokines such as C-Reactive Protein and IL-6. By means of its impact on Nrf2 activation, zinc contributes to wound healing, with an emphasis on hemostasis, inflammation, anti-microbial control, granulation and re-epithelialization [139]. About 30% of zinc is located in bone, and thus is important for bone quality and mineralization. It stimulates bone formation and regeneration by promoting osteoblast proliferation and preventing their apoptosis [140].

Diabetic patients, in particular, require an adequate level of zinc to ensure Nrf2 activation, which protects them against diabetes-induced oxidative damage to their bones and several other organs [141,142].

The recommended daily intake of zinc is 10–15 mg/day, and supplementation is essential in elderly patients, among whom 30% are deficient [143].


**
*Probiotics*
**


These substances are defined by the US Food and Agriculture Organization as “live microorganisms which when administered in adequate amounts confer a health benefit on the host”. Probiotics have demonstrated immuno-protective functions; they have been reported to tighten gut-cell junctions and to decrease rates of antigen presentation on gut epithelial layers, thus affecting intestinal immune cell activation [144].

In 2015, Ohlsson and Sjogren introduced the notion of osteomicrobiology [145]. This science highlights how microbiota regulate skeletal maturation, bone aging and pathological bone loss. Recent publications have reported that probiotics reduce inflammatory cytokines production and have a protective effect on bone [146,147]. Specific probiotics may activate Nrf2 and the expression of antioxidant-related genes. They could prevent bone loss that is associated with periodontitis and peri-implantitis and improve the maintenance of implants over time [148].


**Other nutraceuticals with antioxidant properties**


Vitamins K2, E and B6, copper, magnesium, fatty acids and selenium all have anti-inflammatory and antioxidant functions [149,150,151,152,153,154,155,156,157,158,159]. Their supplementation during the healing period after oral surgery should be considered in order to enhance wound healing and bone regeneration.

Micronutrients and nutraceuticals have effects on Nrf2 activation and the immune response, but also exert direct action on the bone matrix and its maintenance overtime. They may enhance peri-implant wound and bone healing and stability, and therefore, the osseointegration of dental implants and long-term success of dental surgery [160]. Their supplementation should start at least seven to ten days before surgery.

**b.** 
**Antioxidation by reduction in levels of serum LDL cholesterol**


Hypercholesterolemia has been identified as a risk factor for the stability of dental implants and must be treated before surgery if LDL serum levels exceed 1.4 g/L [97]. Treatment can be achieved through the administration of a combination of cholesterol-lowering medications (statins, ezetimibe, bile-acid sequestrants, niacin or bempedoic acid) and an improvement in lifestyle, which involves exercise, a low-saturated-fat diet, not smoking, high blood pressure treatment and sugar intake control.

**c.** 
**Autophagy**


The process of cell regeneration was elucidated in 1963 by Christian de Duve, who coined the term ‘autophagy’, which earned him the Nobel Prize in 1974 [161]. Thirty years later, Yoshinori Ohsumi’s work showed similar mechanisms that occurred in the human body and for which he won the Nobel Prize in 2016. Cell dysfunctions have been implicated in various diseases such as cancer, Parkinson’s disease, diabetes and other genetic and neurological illnesses. When fasting and deprived of nutrients, cells undergo autophagy, during which they form unusually large vacuoles that function as “cell dumps” in which unwanted substances are collected and recycled [162].

Autophagy improves both innate and adaptive immune responses [163] and accelerates wound healing through angiogenesis [164]. Its benefits can be harnessed simply and effectively by fasting intermittently for at least 16 h a day (by skipping breakfast, for example).

**d.** 
**Use of immunomodulatory antibiotics**


Azithromycin is a well-known antibiotic from the macrolide family which was discovered in 1980. In 1987, Kudoh was the first to publish information about the immunomodulatory function of macrolides [165]. Azithromycin is now recognized not only for its potent antibiotic properties, but also as a powerful immunomodulatory agent. It is retrieved in high concentrations in phagocytic cells and fibroblasts, where they serve as drug reservoirs within tissue. They enable activity against microorganisms and the transfer of antibiotic substances to phagocytic cells to combat pathogens at infected sites [166].

Aside from its anti-microbial effects, azithromycin exerts anti-inflammatory activities by modulating dendritic cells and CD4+ T-cell functions. Consequently, it may be of therapeutic benefit in various inflammatory disorders and to prevent graft-versus-host disease in patients undergoing stem-cell transplantation [167,168,169]. Furthermore, azithromycin has a therapeutic effect on diabetes as it inhibits oxidative stress, inflammation and apoptosis [170]. It combats epithelial barrier dysfunctions [171,172] and promotes connective tissue remodeling [173].

When administered as a pre-operative prophylactic antibiotic, a single dose of azithromycin has been found to generate higher concentrations of active substances in periodontal tissues compared with amoxicillin. It has greater potential to inhibit the expression of inflammatory mediators at peri-implant wound sites than amoxicillin [174].

### 5.2. Local Improvement in Immune Response and Anti-Inflammation Strategies

Surgery should be managed to avoid local oxidation. The following protocols improve local anti-inflammation and thus could enhance the immune response after surgery.

**a.** 
**Growth factors and Platelet-Rich Fibrin**


Growth factors have found routine applications in oral surgery and various medical fields. Platelet-Rich Fibrin (PRF), prepared without anticoagulant, represents the simplest and most efficient protocol, as it combines platelets, leukocytes and a fibrin matrix [175]. After two decades of clinical use of PRF, the literature contains substantial evidence of its benefits and its mechanism of action. PRF induces angiogenesis and anti-inflammation pathways [176,177], possesses osteogenic properties [178] and inhibits osteoclastogenesis [179]. Evidence is increasing regarding the anti-inflammation processes that PRF initiates: decrease in the inflammatory response of mesenchymal stem cells and the release of IL-1β from macrophages [180,181,182]. Recent studies indicate that the main mechanism of action of PRF could lies in its antioxidant promotion as it neutralizes the hydrogen peroxide produced in gingival fibroblasts [183] and thus its improvement of local immune responses.

Moreover, PRF enables the production of a sticky bone graft. By reducing the mobility of the graft (induced by muscle activity during smiling, eating, talking or coughing), it avoids fragmentation of the bone callus, which is inflammatory.

In addition, the use of the fibrin matrix enables dentists to perform an open-wound technique: the wound is voluntarily left open and PRF membranes are used to fill the gap between soft-tissue edges to promote the initiation of a secondary wound-healing cascade. PRF acts as a delivery system for growth factors and releases cytokines slowly over at least one week [184,185]. The use of this open-wound technique brings two benefits: it reduces soft-tissue tension at wound closure and increases the width of keratinized mucosa.

**b.** 
**Use of low-inflammatory biomaterials**


Osseointegration is a complex, multifactorial and immune-modulated healing process that involves multiple cells and mediators. This immuno-inflammatory process recruits granulocytes, mesenchymal stem cells and monocytes/macrophages [186,187].

Innate cells continuously survey tissues for foreign substances and have been linked to cell-mediated graft rejection. Their inability to recognize the biomaterial as self stimulates the foreign-body reaction and induces adverse immune reactions, which result in excessive inflammation, impaired healing, fibrotic encapsulation, tissue destruction, or even the isolation and rejection of medical devices. Theses complications suggest a continuous oxidation process, albeit at low levels [188].

The modulation of this response is the key to successful grafts [189]. The foreign-body reaction to biomaterials depends on their biocompatibility, which means that an appropriate biomimetic environment must be created to ensure cell survival [190]. This explains the inflammatory reaction to synthetic material [191], and thus, the growing interest of researchers in organic or inorganic osteogenic materials with antioxidative functions to achieve faster bone repair [192].

Biomaterials can be categorized according to their immunogenicity. Autogenous and allogenous materials have low immunogenicity and express a lower number of inflammation-related genes, and thus, result in low rates of inflammation and the initiation of fewer oxidation processes.

On the other hand, xenogeneic substitutes have good biological activity, but the elimination of immunogenic reactions while retaining osteogenic abilities is a challenge. Indeed, xenoantigenicity represents a major barrier to immune-compatibility [193,194].

Porcine materials, which show close antigenicity to those of humans, can also be considered as a low-inflammatory option. Deproteinized bovine bone (DBBM) expresses a higher number of inflammation-related genes, up to four times more than with allografts [195].

The biggest immunological challenge is to achieve xenoantigen removal from tissues and, at the same time, to yield an intact ECM to serve as a scaffold for bone engineering.

One cause of ECM damage is excessive heating of the biomaterial (over 800–900 °C): the new crystal, far from the original ECM, induces a long-term high level of inflammation [196].

**c.** 
**Pressure-less implant placement**


The objective during implant placement should be to avoid contact between the crestal cortical bone and the implant neck. To achieve this, oral surgeons should either over-drill the cortical bone (with a countersink or larger drill) or place implants subcrestally [197]. New implant designs tend to present narrower necks and to facilitate pressure-less positioning near the crestal bone and improve future Bone-to-Implant Contact (BIC).

Similarly, when surgeons deal with grafted bone upon re-entry, the same attention and the same protocols must be applied. During implant placement in a dense bone graft, torque should be reduced and crestal over-drilling should be performed to avoid graft stress and impaired osseointegration caused by implant pressure in undersized preparation sites [111,198].

**d.** 
**Sutures and tension-free flap closure**


Flap closure should be tension-free on every occasion, in order to avoid ischemia. Flap release is a critical step of the surgery and requires careful consideration to guarantee a tensionless situation. Wound closure and suture techniques should respect the PASS principles, as described by Wang: primary closure, angiogenesis, space creation/maintenance, and stability [199].

**e.** 
**Topical azithromycin**


Azithromycin inhibits human osteoclast function in vitro, leading to a reduction in osteoclast resorptive activity at all concentrations. These findings encourage the use of small doses of azithromycin powder topically before soft-tissue closure, in addition to pre-operative oral antibiotic administration [200].

## 6. Conclusions

The study of osteoimmunology has significantly widened knowledge of bone health and diseases, as it has shed light on the intricate interplay between bone and the immune system. Researchers hope that the interdisciplinary nature of osteoimmunology will lead to major discoveries in bone regeneration and the development of targeted therapies for bone diseases.

Based on the available literature, it could be hypothesized that peri-implantitis is an immune disease, in which bone loss around implants may start with immune deficiency and peri-implant bone oxidation. A lower level of immune stimulation is necessary to maintain bone turnover, and thus, avoid bone resorption. Osteoimmune-supportive protocols, described in this article, include biological patient preparation; the use of azithromycin, PRF and low-inflammatory biomaterials; and the application of specific surgical behaviors such as pressure and tension awareness.

However, a lack of randomized clinical studies and of the application of such protocols in the published literature fails to show yet the benefits of this theoretical approach. This underscores the necessity for further scientific research and validation. Randomized clinical trials that compare the results of current oral surgical protocols with those proposed in this article should highlight the efficiency of clinically applied osteoimmunology. Additional studies on the impact of the combined use of several nutraceuticals on the prevention of dental implants and bone grafts failures could also improve surgical outcomes and patient preparation.

## Figures and Tables

**Figure 1 bioengineering-11-00191-f001:**
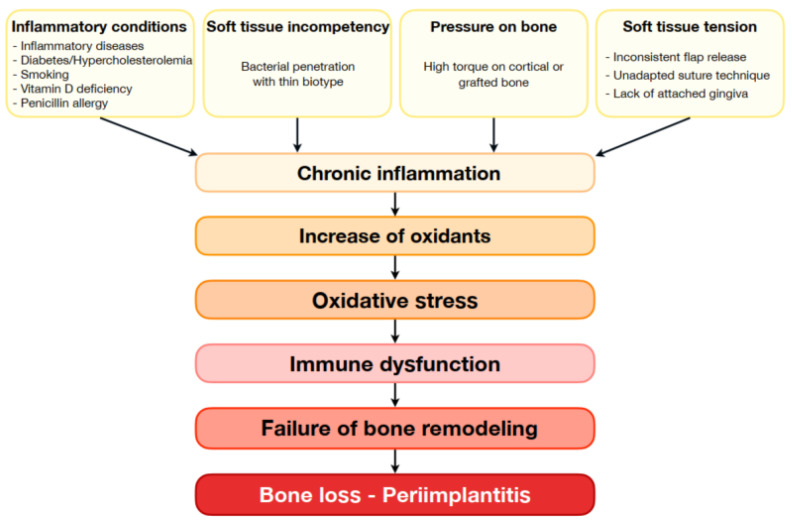
Onset of surgical failures in oral surgery, from inflammatory conditions and clinical features to failure of bone remodeling through oxidative stress and immune dysfunction.

**Figure 2 bioengineering-11-00191-f002:**
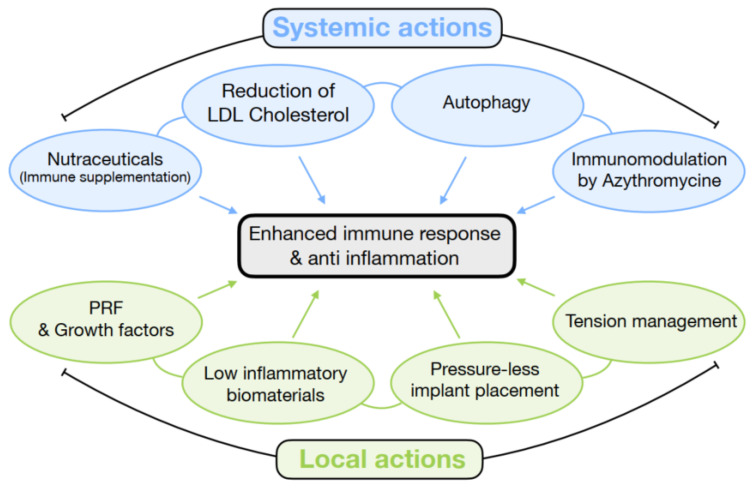
Summary of systemic and local actions to improve immune response and anti-inflammation.

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
