# Peer review of "Bone Formation and Maintenance in Oral Surgery: The Decisive Role of the Immune System—A Narrative Review of Mechanisms and Solutions"

_bioengineering, 2024, doi:10.3390/bioengineering11020191_

Round 1

Reviewer 1 Report

Comments and Suggestions for Authors

The manuscript "Bone Formation and Maintenance in Oral Surgery: The Decisive role of the Immune System. A Narrative Review of Mechanisms and Solutions." present an important theme. But there are some issues that need changed or reviewed before accepted.

Major review:

1) Authors cited Reactive Oxygen Species (ROS), but nothing about the NADPH oxidases (NOX) enzymes, that are responsible for physiological ROS production (Marques-Carvalho et al., 2023: doi: 10.1016/j.bonr.2023.101664). These enzymes (NOX) are involved with bone metabolism (Marques-Carvalho et al., 2023: doi: 10.1016/j.bonr.2023.101664). I am suggesting to authors add that subtopic/item.

2) Some paragraphs no have references. For example:

Pag. 10: “Human bone (either autogenous or allograft) represents a less inflammatory bio-material choice option, and porcine material, which shows close antigenicity to human, can also be considered a low-inflammatory option. Conversely, alloplastic and other xenografts (such as bovine or equine materials) induce long-term inflammation.”

Pags. 10-11: “c. Implant placement

The objective during implant placement is to avoid contact between the crestal cortical bone and the implant neck. To achieve this, oral surgeons should either over-drill the cortical bone (with a countersink or larger drill) or place implants subcrestally. New implant designs tend to present narrower necks and to facilitate pressure-less positioning near the crestal bone.

Similarly, when surgeons deal with grafted bone at re-entry, the same attention and the same protocols must be applied. During implant placement in a dense bone graft, torque should be reduced and crestal over-drilling should be performed, in order to avoid graft stress caused by implant pressure.”

3) About the first example (“Human bone…..induce long-term inflammation”, pag 10), I invite the authors to cite some references about relation the bovine graft and long term inflammation.

Author Response

Dear colleague, 

First of all thank your for your review and detailed answer

1) Indeed NADPH are responsible for ROS production but in our article we wanted to focus on the clinical and biological causes and solutions for bone loss. Redox reactions and oxidative stress is a wide subject and we decided not to dive in the deep fundamental knowledge as the article has more of a comprehensive and clinical aim.

2) We added references as mentionned in your comment

3) This paragraph has been completely modified and references added

Reviewer 2 Report

Comments and Suggestions for Authors

Dear

Thank you for your excellent review. I really learned a lot if evidences and though about some new ideas. However, the title ia not clear to the point of osteoimmunity. I suggest replacing title in a more understandable statement. Some theorise were mentioned as facts in the manuscriot. For example, periimplantitis is an immune disease. This statement needs more evidences before final approval. I also recommend one or two diagram or decision tree figure to calculate and organize data.

Author Response

Dear colleague

Thank your for your review and appreciation of our work.

Indeed, we change the theoritical statement in the conclusion so it appears clearer for readers.

We added figures so the mechanisms and solutions helps organize data.

Reviewer 3 Report

Comments and Suggestions for Authors

TITLE:  Bone Formation and Maintenance in Oral Surgery: The Decisive role of the Immune System. A Narrative Review of Mechanisms and Solutions

The aim of the present investigation was to provide a narrative review regarding the osteoimmunology, and oxidative stress mechanisms involved in bone augmentations failures and peri-implantitis

GENERAL COMMENTS

The article is in-line with the journal topic, but flaws should be improved.  The investigation is interesting, and the present paper is recommended for publication to the present journal after major revision.

The present article seems to reflect a general structure more similar to a book chapter

Abstract:

The structured abstract subtitles are strongly recommended according to the journal guidelines.

The affiliations section should be completed

Introduction

1.     The present section should include a brief classification of peri-implantitis and mucositis as different pathological entities.

2.     In my opinion, the authors should refer to the 2017 Consensus on Peri-implant diseases and conditions.

3.     The relationship between the oxidative stress effect on peri-implantitis is not linear in the introduction section. This aspect should be explained with more details regarding the theoretical pathological pathway cascade process.

2. Osteoimmunology

1.     The section should describe the role of matrix metalloproteinases in human periodontal and peri-implant tissues destruction.

1.     Biological and Metabolic Effects: In my opinion each comorbidity is worthy of a separated analysis due to the different pathological pathways associated with peri-implantitis.

2.     The articles should take in account the reversible and non-plaque-induced marginal bone loss around an osseointegrated implant.

3.     In this way also the calcium homeostasis and chronic kidney disease-mineral and bone disorder could theoretically play a role on peri-implant disease and fixture survival at medium term.

4.     The penicillin allergy paragraph seems to be off-topic considering the aim of the present review. In my opinion it could be removed.

5.     Fig. 1 is missed in this section and should be included in the main text.

6.     The role of nutraceuticals should be investigated also considering the hypothesis of a direct role on bone defect reparation. Recent studies showed in preclinical model an upregulation of the RUNX2, SMAD5, COLL1, COLL4, and COLL5 genes after flavonoids administration (PMID: 24573323; PMID: 36904236).

7.     The following sentence should be expanded and referenced if possible: “Surgery should be managed to avoid local oxidation”

Ethical statements

The following statement are missed.

Author Contributions:

Funding:

Institutional Review Board Statement:

Informed Consent Statement:

Data Availability Statement:

Conflicts of Interest:

Author Response

Dear colleague

Thank you for your review and comments :

1) Abstract has been written according to the editor guideline : "The abstract should be a single paragraph and should follow the style of structured abstracts, but without headings" and thus has no subtitles.

Affilitions section has been completed

2) Definition of mucositis and peri-implantitis have been added in the introduction and a clearer link between oxidation and bone loss around implants.

3) 

We modified and specified kidney diseases in the list of inflammatory and metabloic diseases 

The paragraph on penicillin allergy have been maintened as it described the immune deficiency of these patients and the influence on bone maintenance around implant

Figures have been added to the document

Nutraceuticals paragraph describe some of the direct actions on bone (for vitamine D, C, melatonine, zinc and probiotics). It still has been modified, but this paragraph could represent an entire article and has been made short on purpose so it fits inside our narrative review.

Sentence on local oxidation have been modified.

4) Ethical statement :

Authors contribution and conflicts of interest have been added.

Funding, review, consent and data availability statements are not concern as it is a narrative review, with no clinical cases.

Round 2

Reviewer 1 Report

Comments and Suggestions for Authors

The revised version of the manuscript has been substantially improved. But we suggested double checked:

1) To add a small paragraph about NOXs.

2) To review some paragraphs without references. For example:

“From an osteoimmune point of view, success in dental implants or bone augmentation procedures could depends on the presence of sufficient antioxidants and thus on an effective immune response. Bone loss could be prevented through the management of immunity and oxidation, before, during and after surgery.

Through this narrative review, the authors aspire to guide oral surgeons in the comprehension of osteoimmunology and oxidative stress mechanisms and, based on these principles, to introduce concrete clinical protocols by which to prevent bone augmentations failures and peri-implantitis.”

“Similarly, when surgeons deal with grafted bone at re-entry, the same attention and the same protocols must be applied. During implant placement in a dense bone graft, torque should be reduced and crestal over-drilling should be performed, in order to avoid graft stress caused by implant pressure.”

Author Response

Dear colleague

Thank your for your correction review

  • We added a NOX paragraph and references
  • We added references for bone graft drilling section

For the other paragraph there is no reference as it is a narrative statement and a theory proposal from our group.

Reviewer 3 Report

Comments and Suggestions for Authors

The authors successfully solved the review points and the article is recommended for pubblication in the present form.

Author Response

Dear colleague

Thank you for your review